# Probabilistic Entity Representation Model for Reasoning over Knowledge Graphs

**Nurendra Choudhary[1], Nikhil Rao[2], Sumeet Katariya[2], Karthik Subbian[2], Chandan K. Reddy[1,2]**
[1]Department of Computer Science, Virginia Tech, Arlington, VA
[2]Amazon, Palo Alto, CA
`nurendra@vt.edu`, {`nikhilsr, katsumee, ksubbian`}`@amazon.com`, `reddy@cs.vt.edu`

## Abstract

Logical reasoning over Knowledge Graphs (KGs) is a fundamental technique that can provide efficient querying mechanism over large and incomplete databases. Current approaches employ spatial geometries such as boxes to learn query representations that encompass the answer entities and model the logical operations of projection and intersection. However, their geometry is restrictive and leads to non-smooth strict boundaries, which further results in ambiguous answer entities. Furthermore, previous works propose transformation tricks to handle unions which results in non-closure and, thus, cannot be chained in a stream. In this paper, we propose a Probabilistic Entity Representation Model (PERM) to encode entities as a Multivariate Gaussian density with mean and covariance parameters to capture its semantic position and smooth decision boundary, respectively. Additionally, we also define the closed logical operations of projection, intersection, and union that can be aggregated using an end-to-end objective function. On the logical query reasoning problem, we demonstrate that the proposed PERM significantly outperforms the state-of-the-art methods on various public benchmark KG datasets on standard evaluation metrics. We also evaluate PERM's competence on a COVID-19 drug-repurposing case study and show that our proposed work is able to recommend drugs with substantially better F1 than current methods. Finally, we demonstrate the working of our PERM's query answering process through a low-dimensional visualization of the Gaussian representations.

## 1    Introduction

Knowledge Graphs (KGs) are structured heterogeneous graphs where information is organized as triplets of entity pair and the relation between them. This organization provides a fluid schema with applications in several domains including e-commerce [1], web ontologies [2, 3], and medical research [4, 5]. Chain reasoning is a fundamental problem in KGs, which involves answering a chain of first-order existential (FOE) queries (translation, intersection, and union) using the KGs' relation paths. A myriad of queries can be answered using such logical formulation (some examples are given in Figure 1). Current approaches [6, 7, 8] in the field rely on mapping the entities and relations onto a representational latent space such that the FOE queries can be reduced to mathematical operations in order to further retrieve the relevant answer entities.

Euclidean vectors [6, 9] provide a nice mechanism to encode the semantic position of the entities by leveraging their neighborhood relations. They utilize a fixed threshold over the vector to query for answer entities (such as a k-nearest neighbor search). However, queries differ in their breadth. Certain queries would lead to a greater set of answers than others, e.g., query `Canadians` will result in a higher number of answers than query `Canadian Turing Award winners`. To capture this query behavior, spatial embeddings [7, 8, 10, 11] learn a border parameter that accounts for broadness of

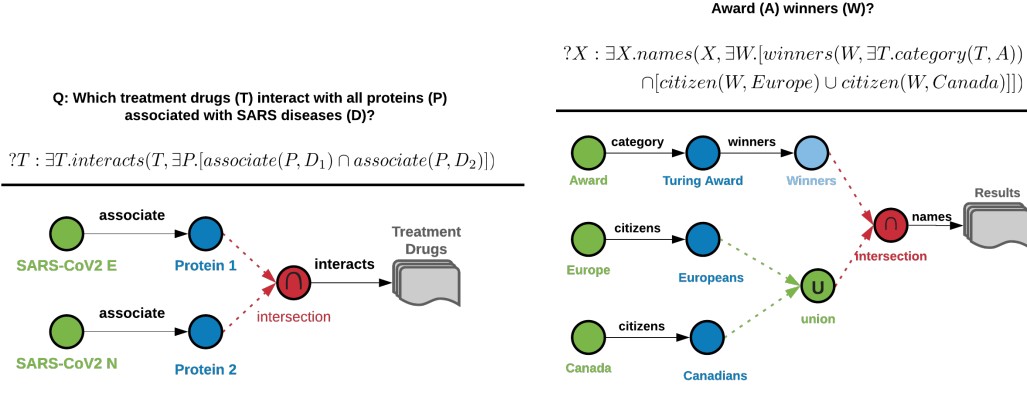

$?X : \exists X.names(X, \exists W.[winners(W, \exists T.category(T, A)) \cap [citizen(W, Europe) \cup citizen(W, Canada)]]])$

Q: Which treatment drugs (T) interact with all proteins (P) associated with SARS diseases (D)?

$?T : \exists T.interacts(T, \exists P.[associate(P, D_1) \cap associate(P, D_2)]])$

(a) Drug Repurposing (DRKG).        (b) Open-domain (FB15K).

Figure 1: Sample FOE queries from different datasets that utilize existential quantification ($\exists$), intersection ($\cap$), and union ($\cup$) operations. The simple operations need to be chained together in an end-to-end objective function to retrieve relevant results for complex queries.

queries by controlling the volume of space enclosed by the query representations. However, these spatial embeddings rely on more complex geometries such as boxes [7] which do not have a closed form solution to the union operation, e.g., the union of two boxes is not a box. Thus, further FOE operations cannot be applied to the union operation. Additionally, their strict borders lead to some ambiguity in the border case scenarios and a non-smooth distance function, e.g., a point on the border will have a much smaller distance if it is considered to be inside the box than if it is considered to be outside. This challenge also applies to other geometric enclosures such as hyperboloids [8].

Another line of work includes the use of structured geometric regions [12, 7] or density functions [13, 14, 11, 15] instead of vector points for representation learning. While these approaches utilize the representations for modeling individual entities and relations between them, we aim to provide a closed form solution to logical queries over KGs using the Gaussian density function which enables chaining the queries together. Another crucial difference in our work is in handling a stream of queries. Previous approaches rely on Disjunctive Normal Form (DNF) transformation which requires the entire query input. In our model, every operation is closed in the Gaussian space and, thus, operations of a large query can be handled individually and aggregated together for the final answers.

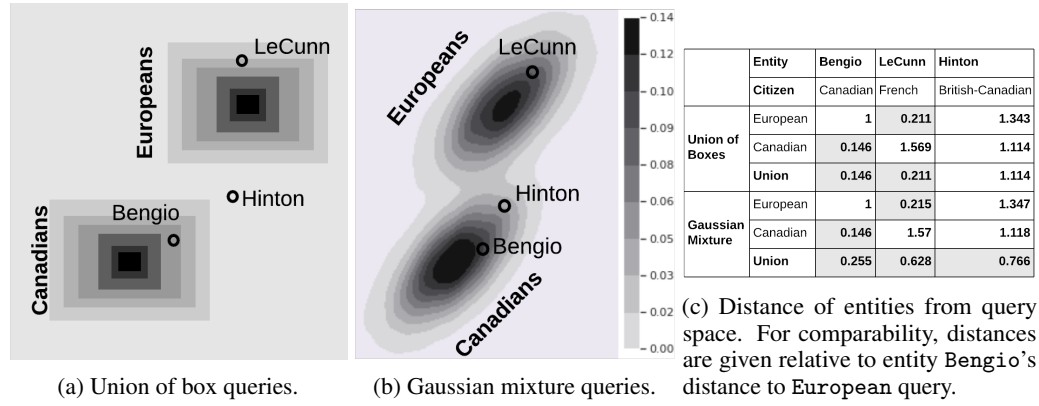

(a) Union of box queries.    (b) Gaussian mixture queries.

(c) Distance of entities from query space. For comparability, distances are given relative to entity `Bengio`'s distance to `European` query.

Figure 2: Results of the query `Europeans` $\cup$ `Canadians`. Entities in the darker areas have higher probability of being the answers than lighter areas. We can observe from (c) that the non-smooth borders of box geometry do not encompass the answer `Hinton`.

To alleviate the drawbacks of operations not being closed under unions and border ambiguities, we propose Probabilistic Entity Representation Model (PERM). PERM models entities as a mixture of Gaussian densities. Gaussian densities have been previously used in natural language processing [14] and graphs [15] to enable more expressive parameterization of decision boundaries. In our case, we

utilize a mixture of multivariate Gaussian densities due to their intuitive closed form solution for translation, intersection, and union operations. In addition, they can also enable the use of a smooth distance function; Mahalanobis distance [16]. Figure 2 provides an example of such a case where the non-smooth boundaries of box query embeddings are not able to capture certain answers. We utilize the mean ($\mu$) and co-variance ($\Sigma$) parameters of multivariate Gaussian densities to encode the semantic position and spatial query area of an entity, respectively. The closed form solution for the operations allows us to solve complex queries by chaining them in a pipeline. PERM does not need to rely on DNF transformations, since all the outputs are closed in the Gaussian space and complex queries can be consolidated in an end-to-end objective function, e.g., in Figure 2b, `Europeans` $\cup$ `Canadians` is a Gaussian mixture and the single objective is to minimize the distance between the mixture and entity `Hinton`, whereas in the case of boxes (shown in Figure 2a), we have two independent objectives to minimize the distance from each box in the union query. Summarizing, the contributions of our work is as follows:

1. We develop Probabilistic Entity Representation Model (PERM), a method to reason over KGs using (mixture of) Gaussian densities. Gaussians are able to provide a closed form solution to intersection and union, and also a smooth distance function. This enables us to process a chain of complex logical queries in an end-to-end objective function.

2. PERM is able to outperform the current state-of-the-art baselines on logical query reasoning over standard benchmark datasets. Additionally, it is also able to provide better drug recommendations for COVID-19.

3. PERM is also interpretable since the Gaussian embeddings can be visualized after each query process to understand the complete query representation.

The rest of the paper is organized as follows: Section 2 presents the current work in the field. In section 3, we present PERM and define its various operations. Section 4 provides the formulation for building the reasoning chains for complex queries. We provide the experimental setup and results in section 5. We conclude our paper in section 6 and present its broader impact in section 7.

## 2   Related Work

The topic of multi-hop chain reasoning over KGs has gained a lot of attention in recent years [17, 18, 19, 6]. These approaches utilize vector spaces to model query representation and retrieve results using a fixed threshold. While such representations are efficient at encoding semantic information, the fixed thresholds that are typically used in these models do not allow for an expressive (adjustable) boundary and, thus, are not best suited for representing queries. Spatial embeddings [7, 8, 20] enhance the simple vector representations by adding a learnable border parameter that controls the spatial area around a query representation. These methods have strict borders that rely on non-smooth distance function that creates ambiguity between border cases. On the other hand, in our model, the variance parameter of the query's Gaussian densities creates soft smoothly increasing borders in terms of the Mahalanobis distance. Additionally, the previous methods do not provide a closed form solution for unions which we solve using Gaussian mixture models.

Density-based embeddings have seen a recent surge of interest in various domains. Word2Gauss [14] provides a method of learning Gaussian densities for words from their distributional semantic information. In addition, the authors further apply this work to knowledge graphs [13]. Another approach [15] aims to learn Gaussian graph representations from their network connections. These methods are, however, focused on learning semantic information and do not easily extend to logical queries over knowledge graphs. PERM primarily focuses on learning spatial Gaussian densities for queries, while also capturing the semantic information. To achieve this, we derive closed form solutions to FOE queries.

## 3   Probabilistic Entity Representation Model for Logical Operators

Knowledge Graphs (KG) $\mathcal{G} : E \times R$ are heterogeneous graphs that store entities ($E$) and relations ($R$). Each relation $r \in R$ is a Boolean function $r : E \times E \to \{True, False\}$ that indicates if the relation $r$ exists between a pair of entities. Without loss of generality, KGs can also be organized as a set of triples $\langle e_1, r, e_2 \rangle \subseteq \mathcal{G}$, defined by the Boolean relation function $r(e_1, e_2)$. In this work, we

focus on the following three FOE operations: translation (t), intersection ($\cap$), and union ($\cup$). The operations are defined as below:

$$q_t[Q_t] \triangleq ?V_t : \{v_1, v_2, ..., v_k\} \subseteq E \, \exists \, a_1 \tag{1}$$

$$q_\cap[Q_\cap] \triangleq ?V_\cap : \{v_1, v_2, ..., v_k\} \subseteq E \, \exists \, a_1 \cap a_2 \cap ... \cap a_i \tag{2}$$

$$q_\cup[Q_\cup] \triangleq ?V_\cup : \{v_1, v_2, ..., v_k\} \subseteq E \, \exists \, a_1 \cup a_2 \cup ... \cup a_i \tag{3}$$

$$\text{where } Q_t = (e_1, r_1); \; Q_\cap, Q_\cup = \{(e_1, r_1), (e_2, r_2), ..(e_i, r_i)\} \text{ and } a_i = r_i(e_i, v_a)$$

where $q_t$, $q_\cap$, and $q_\cup$ are the translation, intersection, and union queries, respectively; and $V_t$, $V_\cap$, and $V_\cup$ are the corresponding results [10]. As we notice above, each entity has a dual nature; one as being part of a query and another as a candidate answer to a query. In PERM, we model the query space of an entity $e_i \in E$ as a multivariate Gaussian density function; $e_i = \mathcal{N}(\mu_i, \Sigma_i)$, where the learnable parameters $\mu_i$ (mean) and $\Sigma_i$ (covariance) indicate the semantic position and the surrounding query density of the entity, respectively. As a candidate, we only consider the $\mu_i$ and ignore the $\Sigma_i$ of the entity. We define the distance of a candidate entity $v_i = \mathcal{N}(\mu_i, \Sigma_i)$ from a query Gaussian $e_j = \mathcal{N}(\mu_j, \Sigma_j)$ using the Mahalanobis distance [16] given by:

$$d_\mathcal{N}(v_i, e_j) = (\mu_j - \mu_i)^T \Sigma_j^{-1} (\mu_j - \mu_i) \tag{4}$$

Additionally, we need to define the FOE operations for the proposed Probabilistic Entity Representation Model. A visual interpretation of the operations; translation, intersection, and union is shown in Figure 3. The operations are defined as follows:

**Translation (t).** Each entity $e \in E$ and $r \in R$ are encoded as $\mathcal{N}(\mu_e, \Sigma_e)$ and $\mathcal{N}(\mu_r, \Sigma_r)$, respectively. We define the translation query representation of an entity $e$ with relation $r$ as $q_t$ and the distance of resultant entity $v_t \in V_t$ from the query as $d_t^q$ given by:

$$q_t = \mathcal{N}(\mu_e + \mu_r, (\Sigma_e^{-1} + \Sigma_r^{-1})^{-1}); \quad d_t^q = d_\mathcal{N}(v_t, q_t) \tag{5}$$

**Intersection ($\cap$).** Intuitively, the intersection of two Gaussian densities implies a random variable that belongs to both the densities. Given that the entity densities are independent of each other, we define the intersection of two entity density functions $e_1, e_2$ as $q_\cap$ and distance of resultant entity $v_\cap \in V_\cap$ from the query as $d_\cap^q$ given by:

$$q_\cap = \mathcal{N}(\mu_{e_1}, \Sigma_{e_1})\mathcal{N}(\mu_{e_2}, \Sigma_{e_2}) = \mathcal{N}(\mu_3, \Sigma_3); \quad d_\cap^q = d_\mathcal{N}(v_\cap, q_\cap) \tag{6}$$

$$\text{where, } \Sigma_3^{-1} = \Sigma_1^{-1} + \Sigma_2^{-1}$$

$$\text{and } \mu_3 = \Sigma_3(\Sigma_2^{-1}\mu_1 + \Sigma_1^{-1}\mu_2) \implies \Sigma_3^{-1}\mu_3 = \Sigma_2^{-1}\mu_1 + \Sigma_1^{-1}\mu_2$$

We provide a brief sketch of the proof that the intersection of Gaussian density functions is a closed operation. A complete proof is provided in Appendix A. Let us consider two Gaussian PDFs $P(\theta_1) = \mathcal{N}(\mu_1, \Sigma_1)$ and $P(\theta_2) = \mathcal{N}(\mu_2, \Sigma_2)$. Their intersection implies a random variable that is distributed as the product, $P(\theta_1)P(\theta_2)$ The intersection $P(\theta) = \mathcal{N}(\mu_3, \Sigma_3)$ is derived as follows:

$$P(\theta) = P(\theta_1).P(\theta_2)$$

$$\log(P(\theta)) = (x - \mu_1)^T \Sigma_1^{-1}(x - \mu_1) + (x - \mu_2)^T \Sigma_2^{-1}(x - \mu_2)$$

$$(x - \mu_3)^T \Sigma_3^{-1}(x - \mu_3) = (x - \mu_1)^T \Sigma_1^{-1}(x - \mu_1) + (x - \mu_2)^T \Sigma_2^{-1}(x - \mu_2)$$

$$\text{Comparing coefficients;} \quad \Sigma_3^{-1} = \Sigma_1^{-1} + \Sigma_2^{-1}; \quad \mu_3 = \Sigma_3(\Sigma_2^{-1}\mu_1 + \Sigma_1^{-1}\mu_2)$$

**Union ($\cup$).** We model the union of multiple entities using Gaussian mixtures. The union of entity density functions given by $e_1, e_2, e_3, ..., e_n$ is defined as $q_\cup$ and the distance of resultant entity $v_\cup \in V_\cup$ from the query as $d_\cup^q$ given by:

$$q_\cup = \sum_{i=1}^n \phi_i \mathcal{N}(\mu_{e_i}, \Sigma_{e_i}); \quad d_\cup^q = \sum_{i=1}^n \phi_i d_\mathcal{N}(v_\cup, \mathcal{N}(\mu_{e_i}, \Sigma_{e_i})) \tag{7}$$

$$\text{where, } \phi_i = \frac{exp\left(\mathcal{N}(\mu_{e_i}, \Sigma_{e_i})\right)}{\sum_{j=1}^n exp\left(\mathcal{N}(\mu_{e_j}, \Sigma_{e_j})\right)}$$

$\phi_i \in \Phi$ are the weights for each Gaussian density in the Gaussian mixture, calculated using the self-attention mechanism over the parameters of the Gaussians in the mixture, i.e., $\mu_{e_i}, \Sigma_{e_i} \forall i : 1 \to n$.

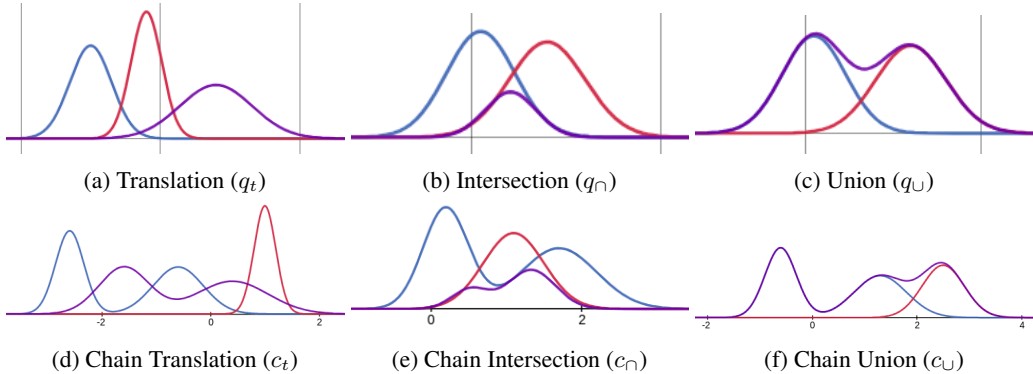

(a) Translation ($q_t$)        (b) Intersection ($q_\cap$)        (c) Union ($q_\cup$)

(d) Chain Translation ($c_t$)      (e) Chain Intersection ($c_\cap$)      (f) Chain Union ($c_\cup$)

Figure 3: The logical single (top row) and chain operations (bottom row) of translation, intersection, and union in the Gaussian space. The operations are closed and will result in either a Gaussian density or a Gaussian mixture. The input operands are given in blue and red and the resultant Gaussian density/mixture is depicted in purple. For simplicity, the example is given for a univariate Gaussian model, but in our work, we use multivariate Gaussian densities.

## 4 Chain Reasoning over Knowledge Graphs

We consider the Gaussian density function (embedding of a single entity) as a special case of Gaussian mixture with a single component. This ensures that all the operations defined in Section 3 are closed under the Gaussian space with an output that is either a single (for translations and intersections) or multi-component Gaussian mixture (for unions). Hence, for chaining the queries, we need to define the logical operators with a Gaussian density and a Gaussian mixture input. In this section, we define the different operators (depicted in Figure 3), in the case of a Gaussian mixture input.

**Chain Translation.** Let us assume that the input query embedding is an $n$-component mixture $p = \sum_{i=1}^{n} \mathcal{N}(\mu_i, \Sigma_i)$ and we need to translate it with relation $r = \mathcal{N}(\mu_r, \Sigma_r)$. Intuitively, we would like to translate all the Gaussians in the mixture with the relation. Hence, we model this translation as $c_t$ and the distance from entities $v_t \in V_t$ as $d_t^c$ given by:

$$c_t = \sum_{i=1}^{n} \phi_i \mathcal{N}(\mu_i + \mu_r, (\Sigma_i^{-1} + \Sigma_r^{-1})^{-1}) \tag{8}$$

$$d_t^c = \sum_{i=1}^{n} \phi_i d_{\mathcal{N}}(v_t, \mathcal{N}(\mu_i + \mu_r, (\Sigma_i^{-1} + \Sigma_r^{-1})^{-1})) \tag{9}$$

**Chain Intersection.** A Gaussian mixture is a union over individual densities. Based on the distributive law of sets, an intersection over a Gaussian mixture $p = \sum_{i=1}^{n} \mathcal{N}(\mu_i, \Sigma_i)$ and entity $e = \mathcal{N}(\mu_e, \Sigma_e)$ implies the union of the intersection between the entity and each Gaussian density in the mixture. Hence, we derive this intersection as $c_\cap$ and the distance from entities $v_\cap \in V_\cap$ as $d_\cap^c$:

$$c_\cap = \cup_{i=1}^{n} \mathcal{N}(\mu_e, \Sigma_e) \mathcal{N}(\mu_i, \Sigma_i) = \sum_{i=1}^{n} \phi_i \mathcal{N}(\mu_e, \Sigma_e) \mathcal{N}(\mu_i, \Sigma_i)$$

$$\implies c_\cap = \sum_{i=1}^{n} \phi_i \mathcal{N}(\mu_{e \cap i}, \Sigma_{e \cap i}) \tag{10}$$

where, $\Sigma_{e \cap i}^{-1} = \Sigma_e^{-1} + \Sigma_i^{-1}$ and $\mu_{e \cap i} = \Sigma_{e \cap i}(\Sigma_i^{-1} \mu_e + \Sigma_e^{-1} \mu_i)$

$$d_\cap^c = \sum_{i=1}^{n} \phi_i d_{\mathcal{N}}(v_\cap, \mathcal{N}(\mu_{e \cap i}, \Sigma_{e \cap i})) \tag{11}$$

**Chain Union.** The union of an entity $e = \mathcal{N}(\mu_e, \Sigma_e)$ with a Gaussian mixture $\sum_{i=1}^{n} \phi_i \mathcal{N}(\mu_i, \Sigma_i)$ is the addition of the entity to the mixture. Hence, the union $c_\cup$ and the distance from entities $v_\cup \in V_\cup$

$d_\cup^c$ can be defined as follows:

$$c_\cup = \sum_{i=1}^{n} \phi_i \mathcal{N}(\mu_i, \Sigma_i) + \phi_e \mathcal{N}(\mu_e, \Sigma_e) \tag{12}$$

$$d_\cup^c = \sum_{i=1}^{n} \phi_i d_\mathcal{N}(v_\cup, \mathcal{N}(\mu_i, \Sigma_i)) + \phi_e d_\mathcal{N}(v_\cup, \mathcal{N}(\mu_e, \Sigma_e)) \tag{13}$$

**Implementation Details.** To calculate the weights ($\phi_i \in \Phi$) of the Gaussian mixtures, we use the popular self-attention mechanism [21]. The gradient descent over Mahalanobis distance (Eq. 4) and derivation for the product of Gaussians (Eq. 6) are given by [22] and Appendix A, respectively. Another important note is that we do not need to compute $\Sigma$ for the operations, but rather we only need to compute the $\Sigma^{-1}$. Also, storing the complete $\Sigma^{-1}$ requires quadratic memory, i.e., a Gaussian density of $d$ variables requires $d \times d$ parameters for $\Sigma$. So, we only store a decomposed matrix $L$ of $\Sigma^{-1}: \Sigma^{-1} = LL^T$. Thus, for a Gaussian density of $d$ variables our memory requirement is $d \times (r+1)$ parameters ($d$ for $\mu$ and $d \times r$ for $\Sigma^{-1}$). For computing the $\mu_3$ for intersection, in Eq. (6), we use a linear solver (`torch.solve`) for faster computation. All our models are implemented in Pytorch [23] and run on four Quadro RTX 8000. [1]

## 5 Experiments

This section describes the experimental setup used to analyze the performance of PERM on various tasks with a focus on the following research questions:

1. Does PERM's query representations perform better than the state-of-the-art baselines on the task of logical reasoning over standard benchmark knowledge graphs?

2. What is the role of individual components in PERM's overall performance gain?

3. Is PERM able to recommend better therapeutic drugs for COVID-19 from drug re-purposing graph data compared to the current baselines?

4. Are we able to visualize the operations on PERM's query representations in the latent space?

### 5.1 Datasets and Baselines

We utilize the following standard benchmark datasets to compare PERM's performance on the task of reasoning over KGs:

- **FB15K-237** [24] is comprised of the 149,689 relation triples and textual mentions of Freebase entity pairs. All the simply invertible relations are removed.

- **NELL995** [25] consists of 107,982 triples obtained from the $995^{th}$ iteration of the Never-Ending Language Learning (NELL) system.

- **DBPedia**[2] is a subset of the Wikipedia snapshot that consists of a multi-level hierarchical taxonomy over 240,942 articles.

- **DRKG** [26] (Drug Re-purposing Knowledge Graph) is used to evaluate the performance of our model on both the logical reasoning and drug recommendation tasks.

Table 1: Dataset statistics including the number of unique entities, relations, and edges, along with the splits of dataset triples used in the experiments.

| Dataset | # Entities | # Relations | # Edges | # Training | # Validation | # Test |
|---------|-----------|-------------|---------|-----------|--------------|--------|
| FB15k-237 | 14,505 | 237 | 310,079 | 272,115 | 17,526 | 20,438 |
| NELL995 | 63,361 | 200 | 142,804 | 114,213 | 14,324 | 14,267 |
| DBPedia | 34,575 | 3 | 240,942 | 168,659 | 24,095 | 48,188 |
| DRKG | 97,238 | 107 | 5,874,271 | 4,111,989 | 587,428 | 1,174,854 |

---

[1]Implementation code: `https://github.com/Akirato/PERM-GaussianKG`

[2]`https://www.kaggle.com/danofer/dbpedia-classes`

More detailed statistics of these datasets are provided in Table 1. For our experiments, we select the following baselines based on (i) their performance on the logical reasoning task and (ii) their ability to extend to all FOE query combinations.

- **Graph Query Embedding (GQE)** [6] embeds entities and relations as a vector and utilizes TransE [17] to learn the query embeddings. The distance of the answer entities is calculated using L1-norm.

- **Query2Box (Q2B)** [7] embeds entities and relations as axis aligned hyper-rectangles or boxes and utilize FOE queries to learn query representations. The distance of answer entities is given by a weighted combination of the answer's distance from the center and the border of the query box.

- **Beta Query Embedding (BQE)** [11] utilizes beta distribution to learn query representations from FOE queries with a novel addition of negation queries. The distance is calculated as the dimension-wise KL divergence between the answer entity and the query beta embedding.

- **Complex Query Decomposition (CQD)** [10] answers complex queries by reducing them to simpler sub-queries and aggregating the resultant scores with t-norms.

Some of the other baselines [27, 18] focus solely on the multi-hop problem. They could not be intuitively extended to handle all FOE queries, and hence, we did not include them in our study.

Table 2: Performance comparison of PERM against the baselines to study the efficacy of the query representations. The columns present the different query structures and the overall average performance. The last row presents the Average Relative Improvement (%) of PERM compared to CQD over all datasets across different query types. Best results for each dataset are shown in bold. The MRR results for experiments are given in Appendix C.

| | | HITS@3 | | | | | | | | | |
|---|---|---|---|---|---|---|---|---|---|---|---|
| **Dataset** | **Model** | **1t** | **2t** | **3t** | **2∩** | **3∩** | **2∪** | **∩t** | **t∩** | **∪t** | **Avg** |
| FB15k-237 | GQE | .404 | .214 | .147 | .262 | .390 | .164 | .087 | .162 | .155 | .221 |
| | BQE | .455 | .122 | .102 | .232 | .459 | .141 | .224 | .124 | .101 | .218 |
| | Q2B | .467 | .240 | .186 | .324 | .453 | .239 | .050 | .108 | .193 | .251 |
| | CQD | .512 | **.288** | **.221** | .352 | .457 | .284 | **.129** | **.249** | .121 | .290 |
| | PERM | **.520** | .286 | .216 | **.361** | **.490** | **.305** | .128 | .212 | **.239** | **.306** |
| NELL995 | GQE | .417 | .231 | .203 | .318 | .454 | .200 | .081 | .188 | .139 | .248 |
| | BQE | **.711** | .156 | .132 | **.438** | .540 | .153 | **.250** | .160 | .091 | .292 |
| | Q2B | .555 | .266 | .233 | .343 | .480 | .369 | .132 | .212 | .163 | .306 |
| | CQD | .667 | **.350** | **.288** | .410 | **.529** | **.531** | .171 | **.277** | .156 | **.375** |
| | PERM | .581 | .286 | .243 | .352 | .508 | .460 | .143 | .195 | **.200** | .328 |
| DBPedia | GQE | .673 | .006[3] | N.A. | .873 | .879 | .402 | .160 | .668 | 0.00 | .458 |
| | BQE | .881 | **.007**[3] | N.A. | **1.00** | **1.00** | .384 | **.435** | .590 | 0.00 | .565 |
| | Q2B | .832 | **.007**[3] | N.A. | **1.00** | **1.00** | .649 | .224 | .856 | 0.00 | .571 |
| | CQD | .870 | **.007**[3] | N.A. | **1.00** | **1.00** | .673 | .218 | .787 | 0.00 | .569 |
| | PERM | **.950** | **.007**[3] | N.A. | **1.00** | **1.00** | **.782** | .232 | **.952** | 0.00 | **.615** |
| DRKG | GQE | .420 | .218 | .153 | .270 | .409 | .181 | .101 | .186 | .174 | .235 |
| | BQE | .554 | .141 | .123 | .347 | .512 | .185 | **.281** | .173 | .124 | .271 |
| | Q2B | .499 | .263 | .199 | .337 | .489 | .284 | .068 | .134 | .235 | .279 |
| | CQD | .554 | **.323** | **.238** | .369 | .495 | .341 | .184 | **.310** | .150 | .329 |
| | PERM | **.565** | .322 | .236 | **.387** | **.540** | **.376** | .190 | .273 | **.297** | **.354** |
| **PERM vs Q2B (%)** | | 10.9 | 12.3 | 13.0 | 7.20 | 6.10 | 26.3 | 84.2 | 50.8 | 24.3 | 15.9 |
| **PERM vs CQD (%)** | | 3.80 | -0.9 | -2.4 | 2.00 | 5.80 | 9.50 | 1.80 | -5.5 | 93.0 | 6.2 |

## 5.2 (RQ1) Reasoning over KGs

To evaluate the efficacy of PERM's query representations, we compare it against the baselines on different FOE query types; (i) Single Operator: $1t$, $2t$, $3t$, $2\cap$, $3\cap$, $2\cup$ and (ii) Compound Queries: $\cap t$, $t\cap$, $\cup t$. We follow the standard evaluation protocol [7, 11, 8] and utilize the three splits of a KG

---

[3]DBPedia has an extremely large number of resultant grand-children leaves ($\approx 10^3$ per grand-parent) for the 2t task and, thus, we notice poor performance on 2t task across all the evaluation models.

for training $\mathcal{G}_{train}$, validation $\mathcal{G}_{valid}$, and evaluation $\mathcal{G}_{test}$ (details in Table 1). The models are trained on $\mathcal{G}_{train}$ with validation on $\mathcal{G}_{valid}$. The final evaluation metrics for comparison are calculated on $\mathcal{G}_{test}$. For the baselines, we calculate the relevance of the answer entities to the queries based on the distance measures proposed in their respective papers. In PERM, the distance of the answer entity from the query Gaussian density is computed according to the measures discussed in Sections 3 and 4. We use the evaluation metrics of HITS@K and MRR to compare the ranked set of results obtained from different models. Given the ground truth $\hat{E}$ and model outputs $\{e_1, e_2, ..., e_n\} \in E$, the metrics are calculated as follows:

$$\text{HITS@K} = \frac{1}{K} \sum_{k=1}^{K} f(e_k); \ f(e_k) = \begin{cases} 1, \text{if } e_k \in \hat{E} \\ 0, \text{ else} \end{cases}$$

$$\text{MRR} = \frac{1}{n} \sum_{i=1}^{n} \frac{1}{f(e_i)}; \ f(e_i) = \begin{cases} i, \text{if } e_i \in \hat{E} \\ \infty, \text{ else} \end{cases}$$

From the results provided in Table 2, we observe that PERM, is able to outperform all the current state-of-the-art approaches, on an average across all FOE queries by 6.2%. Specifically, we see a consistent improvement for union queries; 9.5% and 93% in the case of 2∪ and ∪$t$, respectively. Comparing the models based on only geometries, we notice the clear efficacy of PERM query representations with an average improvement of 37.9%, 15.9%, and 37.3% over vectors (GQE), boxes (Q2B), and beta distribution (BQE), respectively. Given these improvements and the ability to handle compound queries in an end-to-end manner, we conclude that Gaussian distributions are better at learning query representations for FOE reasoning over KGs. Additionally, we provide PERM's results on sample queries from different datasets in Table 3.

Table 3: Qualitative results of PERM on samples from different datasets. Results given in green and red indicate a correct and incorrect prediction, respectively.

| Query | Results |
|---|---|
| Who are European and Canadian Turing awards winners? | Jeffrey Hinton, Yoshua Bengio, Andrew Yao |
| Which Actors and Football Players also became Governors? | Arnold Schwarzenegger, Heath Shuler, Frank White |
| Which treatment drugs interact with all proteins associated with SARS diseases? | Ribavirin, Dexamethasone, Hydroxychloroquine |

## 5.3 (RQ2) Ablation Study

In this section, we evaluate the need for different components and their effects on the overall performance of our model. First, we look at the contribution of utilizing different types of queries to the performance of our model. For this, we train our model on different subsets of queries; (i) only 1$t$ queries, (ii) only translation (1$t$,2$t$,3$t$) queries and (iii) only single operator queries (1$t$,2$t$,3$t$,2∩,3∩,2∪). Furthermore, we look at the need for attentive aggregation in the case of union of Gaussian mixtures. We test other methods of aggregation; (i) vanilla averaging and (ii) MLP [28].

Table 4: Ablation study results. Performance comparison of PERM (final) against different variants of our model. *1t*, *translation* and *single* utilize the 1-hop queries, all translation queries and all single operator queries, respectively. The *average* and *MLP* variants utilize vanilla averaging and MLP for aggregation in union queries. The metrics reported here are an average over all the datasets. Finer evaluation with results for each dataset is given in Appendix D. Best results are given in bold.

| Model Variants | HITS@3 | | | | | | | | | |
|---|---|---|---|---|---|---|---|---|---|---|
| | 1t | 2t | 3t | 2∩ | 3∩ | 2∪ | ∩t | t∩ | ∪t | Avg |
| PERM-1$t$ | .649 | .141 | .128 | .410 | .466 | .477 | .095 | .257 | .102 | .303 |
| PERM-translation | .649 | .182 | .179 | .463 | .535 | .479 | .128 | .308 | .143 | .341 |
| PERM-single | .652 | **.225** | .228 | .524 | .632 | .475 | .167 | .398 | .181 | .387 |
| PERM-average | .628 | .222 | .224 | .524 | .624 | .444 | .158 | .387 | .180 | .377 |
| PERM-MLP | .642 | **.225** | .228 | **.526** | .631 | .462 | .166 | .400 | .183 | .385 |
| PERM (final) | **.654** | **.225** | **.232** | .525 | **.635** | **.481** | **.170** | **.408** | **.184** | **.390** |

From Table 4, we notice that utilizing only $1t$ queries significantly reduces the performance of our model by 22.3% and even increasing the scope to all translation queries is still lower in performance by 12.5% for this case. However, we notice that training on all single operator queries results in comparable performance to the final PERM model. But, given the better overall performance, we utilize all the queries in our final model. For union aggregation, we observe that attention has a clear advantage and both vanilla averaging and MLP lead to a lower performance by 3.33% and 1.28%, respectively. Thus, we adopt self-attention in our final model.

## 5.4 (RQ3) Case Study: Drug Recommendation

In this experiment, we utilize the expressive power of PERM's query representations to recommend therapeutic drugs for COVID-19 from the DRKG dataset. Drugs in the dataset are already approved for other diseases and the aim is to utilize the drug-protein-disease networks and employ them towards treating COVID-19. This can potentially reduce both the drug development time and cost [29]. For this experiment, we utilize the treatment relation in DRKG and retrieve drugs $D : D \xrightarrow{treats} X$, where $X$ is a set of SARS diseases related to the COVID-19 virus. Given that we only need these limited set of entity types (only SARS diseases and drugs) and relation types (only treatments), we only consider the DRKG subgraph that contains this necessary set of entities and relations for learning the representations. We compare the recommendations of different models against a set of actual candidates currently in trials for COVID-19. We use the top-10 recommendations with the evaluation metrics of precision, recall, and F1-score for comparison.

Table 5: Performance comparison of various models on the COVID-19 drug recommendation problem using precision (P), recall (R), and F1-score (F1) metrics. The top three drugs recommended by the models are given in the last column. The recommendations given in green and red indicate correct and incorrect predictions, respectively. The last two rows provide the average relative improvement of PERM compared to the state-of-the-art baselines Q2B and CQD.

| Model | P@10 | R@10 | F1 | Top Recommended Drugs |
|---|---|---|---|---|
| GQE | .119 | .174 | .141 | Piclidenoson, Ibuprofen, Chloroquine |
| BQE | .159 | .200 | .177 | Ribavirin,Oseltamivir, Ruxolitinib |
| Q2B | .194 | .255 | .221 | Ribavirin, Dexamethasone, Deferoxamine |
| CQD | .209 | .260 | .232 | Ribavirin, Dexamethasone, Tofacitinib |
| PERM | **.217** | **.269** | **.251** | Ribavirin, Dexamethasone, Hydroxychloroquine |
| **PERM vs Q2B (%)** | 11.9 | 5.5 | 13.6 | |
| **PERM vs CQD (%)** | 3.8 | 3.5 | 8.2 | |

We can observe from Table 5 that PERM is able to provide the best drug recommendations, across all evaluation metrics. Our model is able to outperform the current methods by atleast 3.8%, 3.5%, and 8.2% in precision, recall, and F1, respectively. Also, the top recommended drugs by our PERM are more inline with the current drug development candidates, thus, showing the better performance of our model's query representations.

## 5.5 (RQ4) Visualization of the Gaussian Representations

To visualize the entity and query in the latent space, we extract representative entity samples from the FB15K-237 dataset and present them in a 2-dimensional space for better comprehension.

Figure 4 depicts the different entities and the mechanism through which PERM narrows down to the particular answer set. Notice that, we are able to perform an intersection after a union operation due to the closed form nature of our operations. This is currently not possible in state-of-the-art baseline methods. Additionally, it should be noted that, unions widen the query space and intersections narrow them down (as expected). Furthermore, the variance parameter acts as a control over the spatial area that an entity should cover and more general entities such as `Turing Award` and `Europe` occupy a larger area than their respective sub-categories, namely, `winners` and `Europeans`.

**Q: Who (X) are the Canadian (C) and European (E) Turing (T) Award (A) winners (W)?**

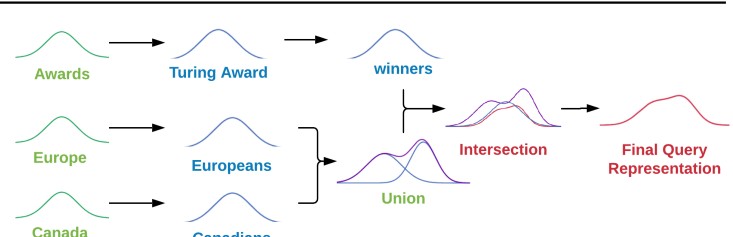
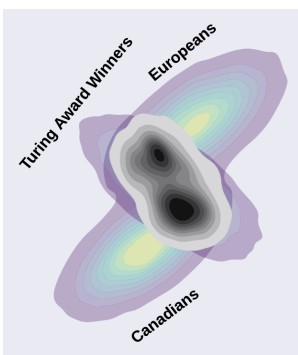

(a) Query processing in PERM. This figure depicts a univariate version of the entity Gaussian embeddings for better visualization of the process. The same property, however, generalizes over an increased number of dimensions, i.e., multivariate case.

(b) Bivariate version of the final query space, given in grayscale with darker colors representing a higher probability of answers.

Figure 4: An illustration of the flow for a sample complex query in the representational space. We note that intersection after union is possible in our PERM model because the operations are closed in Gaussian distributions and this is not possible in current methods including BQE, Q2B, and CQD.

## 6 Conclusion

In this paper, we present Probabilistic Entity Representation Model (PERM), a model to learn query representations for chain reasoning over knowledge graphs. We show the representational power of PERM by defining closed form solutions to FOE queries and their chains. Additionally, we also demonstrate its superior performance compared to its state-of-the-art counterparts on the problems of reasoning over KGs and drug recommendation for COVID-19 from the DRKG dataset. Furthermore, we exhibit its interpretability by depicting the representational space through a sample query processing pipeline.

## 7 Broader Impact

PERM is the first method that models an individual entity in knowledge graphs using Gaussian density function, making it possible to solve FOE queries using a closed form solution. This enables its application in domains that require chain reasoning. The main idea of the proposed solution can also be extended to any domain that can encode its basic units as Gaussians and extend the units through FOE queries, e.g., in topic modeling, topics can be encoded as Gaussians and documents as union of topics.

However, PERM depends on the integrity of the knowledge graph used for training. Any malicious attacks/errors [30, 31] that lead to incorrect relations could, further, lead to incorrect results and affect the confidence of our model. Furthermore, due to the connected nature of complex queries, this attack could propagate and affect a larger set of queries. Such incorrect results would be problematic in sensitive areas of research such as drug recommendations and cybersecurity and, thus, it is necessary to maintain the integrity of training data before learning representations and querying with PERM.

**Acknowledgments**

**Funding:** This work was supported in part by the US National Science Foundation grant IIS-1838730 and Amazon AWS cloud computing credits for research.

**Competing Interests:** All the authors had some form of affiliation with Amazon in the past 36 months.

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
