The following is the supplementary Appendix for the paper; *Probabilistic Entity Representation Model for Reasoning over Knowledge Graphs*. All the references given in the following sections are made in context of the main paper.

# A   Derivation for Product of Multivariate Gaussians

The following sections provide the proof for the product of Gaussians for both the univariate case and multivariate case (used in Eqs.(6) and (10)).

## A.1   Univariate Case

$$\mathcal{N}(\mu, \sigma) = \exp\left(\left(\frac{x-\mu}{\sigma}\right)^2\right)$$

$$P(\theta) = P(\theta_1)P(\theta_2) = \exp\left(\left(\frac{x-\mu_1}{\sigma_1}\right)^2\right).\exp\left(\left(\frac{x-\mu_2}{\sigma_2}\right)^2\right)$$

$$\log P(\theta) = \left(\frac{x-\mu_1}{\sigma_1}\right)^2 + \left(\frac{x-\mu_2}{\sigma_2}\right)^2$$

$$= \frac{(\sigma_2^2 + \sigma_1^2)x^2 - 2(\sigma_1^2\mu_2 + \sigma_2^2\mu_1)x + (\mu_1^2\sigma_2^2 + \mu_2^2\sigma_1^2)}{\sigma_1^2\sigma_2^2}$$

$$= \frac{x^2 - 2\frac{(\sigma_1^2\mu_2 + \sigma_2^2\mu_1)}{\sigma_2^2+\sigma_1^2}x + \frac{\mu_1^2\sigma_2^2 + \mu_2^2\sigma_1^2}{\sigma_2^2+\sigma_1^2}}{\frac{1}{\sigma_1^2} + \frac{1}{\sigma_2^2}}$$

$$= \left(\frac{x - \frac{(\sigma_1^2\mu_2 + \sigma_2^2\mu_1)}{\sigma_2^2+\sigma_1^2}}{(\frac{1}{\sigma_1^2} + \frac{1}{\sigma_2^2})^{-2}}\right)^2 + K, \text{where } K = \frac{\mu_1^2\sigma_2^2 + \mu_2^2\sigma_1^2}{\sigma_1^2\sigma_2^2} - \left(\frac{\sigma_1^2\mu_2 + \sigma_2^2\mu_1}{\sigma_1^2\sigma_2^2}\right)^2$$

$$P(\theta) \propto \exp\left(\left(\frac{x - \frac{(\sigma_1^2\mu_2 + \sigma_2^2\mu_1)}{\sigma_2^2+\sigma_1^2}}{(\frac{1}{\sigma_1^2} + \frac{1}{\sigma_2^2})^{-2}}\right)^2\right) \approx \mathcal{N}\left(\frac{(\sigma_1^2\mu_2 + \sigma_2^2\mu_1)}{\sigma_2^2+\sigma_1^2}, (\frac{1}{\sigma_1^2} + \frac{1}{\sigma_2^2})^{-2}\right)$$

## A.2   Multivariate Case

$$\mathcal{N}(\mu, \Sigma) = \exp\left((x-\mu)^T\Sigma^{-1}(x-\mu)\right)$$

$$P(\theta) = P(\theta_1)P(\theta_2) = \exp\left((x-\mu_1)^T\Sigma_1^{-1}(x-\mu_1)\right).\exp\left((x-\mu_2)^T\Sigma_2^{-1}(x-\mu_2)\right)$$

$$\log(P(\theta)) = (x-\mu_1)^T\Sigma_1^{-1}(x-\mu_1) + (x-\mu_2)^T\Sigma_2^{-1}(x-\mu_2)$$

$$= x^T\Sigma_1^{-1}x - \mu_1^T\Sigma_1^{-1}x - x^T\Sigma_1^{-1}\mu_1 - \mu_1^T\Sigma_1^{-1}\mu_1 + x^T\Sigma_2^{-1}x - \mu_2^T\Sigma_2^{-1}x - x^T\Sigma_2^{-1}\mu_2 - \mu_2^T\Sigma_2^{-1}\mu_2$$

$$= x^T(\Sigma_1^{-1} + \Sigma_2^{-1})x - (\mu_1^T\Sigma_1^{-1} + \mu_2^T\Sigma_2^{-1})x - x^T(\Sigma_1^{-1}\mu_1 + \Sigma_2^{-1}\mu_2) - (\mu_1^T\Sigma_1^{-1}\mu_1 + \mu_2^T\Sigma_2^{-1}\mu_2)$$

Let's assume $P(\theta) \propto \mathcal{N}(\mu_3, \Sigma_3)$, then,

$$log(P(\theta)) = (x-\mu_3)^T\Sigma_3^{-1}(x-\mu_3) + K$$

$$= x^T\Sigma_3^{-1}x - x^T\Sigma_3^{-1}\mu_3 - \mu_3^T\Sigma_3^{-1}x + \mu_3^T\Sigma_3^{-1}\mu_3 + K$$

Comparing coefficients,

$$\Sigma_3^{-1} = \Sigma_1^{-1} + \Sigma_2^{-1}$$

$$\Sigma_3^{-1}\mu_3 = \Sigma_1^{-1}\mu_1 + \Sigma_2^{-1}\mu_2$$

$$\implies \mu_3 = \Sigma_3(\Sigma_1^{-1}\mu_1 + \Sigma_2^{-1}\mu_2)$$

$$\mu_3 = (\Sigma_1^{-1} + \Sigma_2^{-1})^{-1}(\Sigma_1^{-1}\mu_1 + \Sigma_2^{-1}\mu_2)$$

Notice that we need $\Sigma_3$ while calculation $\mu_3$. However, to save computational memory, we only store the inverses of covariances, i.e., $\Sigma_1^{-1}, \Sigma_2^{-1}$ and $\Sigma_3^{-1}$. So, to solve for $\mu_3$ and avoid the computationally

expensive process of matrix inversion, we use the linear solver `torch.solve` on the equation $\Sigma_3^{-1}\mu_3 = \Sigma_1^{-1}\mu_1 + \Sigma_2^{-1}\mu_2$.

## B   Algorithm for KG Reasoning with PERM

Algorithm 1 provides an outline of PERM's overall framework to learn representations of entities $e \in E$ and relations $r \in R$. The algorithm describes the training from FOE operations of translation (lines 4-7), intersection (lines 8-11), and union (lines 12-15).

---

**Algorithm 1:** PERM training algorithm

---

**Input:** Training data $D_t, D_\cap, D_\cup$, which are set of all (query ($Q$), result ($V$)) for translation, intersection, and union, respectively;
**Output:** Entity $E$ and Relation $R$ gaussian density functions;
1 Randomly initialize $e = \mathcal{N}(\mu_e, \Sigma_e) \in E$ and $r = \mathcal{N}(\mu_r, \Sigma_r) \in R)$;
2 **for** *number of epochs; until convergence* **do**
3 $\quad$ $l = 0$; # Initialize loss
4 $\quad$ **for** $\{(e, r, V_t) \in D_t\}$ **do**
5 $\quad\quad$ $q_t = \mathcal{N}(\mu_e + \mu_r, (\Sigma_e^{-1} + \Sigma_r^{-1})^{-1})$ from Eq. (5)
$\quad\quad$ # Update loss for translation queries
6 $\quad\quad$ $l = l + \sum_{v_t \in V_t} d_\mathcal{N}(v_t, q_t)$
7 $\quad$ **end**
8 $\quad$ **for** $\{(Q_\cap, V_\cap) \in D_\cap\}$ **do**
9 $\quad\quad$ $q_\cap = \mathcal{N}(\mu_3, \Sigma_3)$, from Eq. (6)
$\quad\quad$ # Update loss for intersection queries
10 $\quad\quad$ $l = l + \sum_{v_\cap \in V_\cap} d_\mathcal{N}(v_\cap, q_\cap)$
11 $\quad$ **end**
12 $\quad$ **for** $\{(Q_\cup, V_\cup) \in D_\cup\}$ **do**
13 $\quad\quad$ $q_\cup = \sum_{i=1}^n \phi_i \mathcal{N}(\mu_{e_i}, \Sigma_{e_i})$ from Eq. (7)
$\quad\quad$ # Update loss for union queries
14 $\quad\quad$ $l = l + \sum_{v_\cup \in V_\cup} \sum_{i=1}^n \phi_i d_\mathcal{N}(v_\cup, \mathcal{N}(\mu_{e_i}, \Sigma_{e_i}))$
15 $\quad$ **end**
$\quad$ # Update E and R with backpropagation
16 $\quad$ $E \leftarrow E - \Delta_E l$
18 $\quad$ $R \leftarrow R - \Delta_R l$
19 **end**
20 **return** *E,R*

---

## C   MRR metrics for Reasoning over KGs

Table 6 provides the Mean Reciprocal Rank (MRR) results for the reasoning over KGs experiment, given in section 5.

## D   Finer Evaluation of Ablation Study

Table 7 provides finer results of our ablation study.

Table 6: Performance comparison of PERM against the baselines to study the efficacy of the query representations. The columns present the different query structures and the overall average performance. The last two rows presents the Average Relative Improvement (%) of PERM compared to Q2B and CQD over all datasets across query types. Best results for each dataset are shown in bold.

| Metrics | | Mean Reciprocal Rank | | | | | | | | | |
|---|---|---|---|---|---|---|---|---|---|---|---|
| **Dataset** | **Model** | **1t** | **2t** | **3t** | **2∩** | **3∩** | **2∪** | **∩t** | **t∩** | **∪t** | **Avg** |
| FB15k-237 | GQE | .346 | .191 | .144 | .258 | .361 | .144 | .087 | .164 | .149 | .205 |
| | BQE | .390 | .109 | .100 | .228 | **.425** | .124 | .224 | .126 | .097 | .203 |
| | Q2B | .400 | .225 | .173 | .275 | .378 | .198 | .105 | .180 | .178 | .235 |
| | CQD | .439 | **.270** | **.206** | .299 | .381 | .235 | **.271** | **.415** | .112 | .292 |
| | PERM | **.445** | .268 | .201 | **.306** | .409 | **.253** | .269 | .353 | **.220** | **.303** |
| NELL995 | GQE | .311 | .193 | .175 | .273 | .399 | .159 | .078 | .168 | .130 | .210 |
| | BQE | **.530** | .130 | .114 | **.376** | **.475** | .122 | **.241** | .143 | .085 | .246 |
| | Q2B | .413 | .227 | .208 | .288 | .414 | .266 | .125 | .193 | .155 | .254 |
| | CQD | .442 | **.251** | **.226** | .304 | .441 | **.348** | .124 | **.212** | .104 | **.273** |
| | PERM | .432 | .244 | .217 | .296 | .438 | .332 | .122 | .178 | **.190** | .272 |
| DBPedia | GQE | .502 | .005 | N.A. | .749 | .773 | .320 | .154 | .597 | 0.00 | .388 |
| | BQE | .657 | **.006** | N.A. | **.964** | **.966** | .306 | **.419** | .527 | 0.00 | .481 |
| | Q2B | .619 | **.006** | N.A. | .840 | .863 | .468 | .212 | .779 | 0.00 | .473 |
| | CQD | .648 | **.006** | N.A. | .840 | .863 | .485 | .206 | .716 | 0.00 | .471 |
| | PERM | **.706** | **.006** | N.A. | .841 | .862 | **.564** | .219 | **.869** | 0.00 | **.508** |
| DRKG | GQE | .313 | .182 | .132 | .232 | .360 | .144 | .097 | .166 | .163 | .199 |
| | BQE | .413 | .118 | .106 | .298 | .451 | .147 | **.270** | .154 | .116 | .230 |
| | Q2B | .371 | .225 | .178 | .283 | .422 | .205 | .064 | .122 | .223 | .233 |
| | CQD | .413 | **.277** | **.213** | .310 | .427 | .246 | .174 | **.282** | .143 | .276 |
| | PERM | **.420** | .276 | .211 | **.325** | **.465** | **.271** | .179 | .249 | **.282** | **.298** |
| **PERM vs Q2B (%)** | | 11.1 | 16.3 | 12.5 | 4.90 | 4.70 | 24.9 | 55.9 | 29.4 | 24.5 | 20.5 |
| **PERM vs CQD (%)** | | 4.20 | -1.2 | -2.5 | 1.40 | 4.40 | 10.6 | 1.80 | 1.50 | 92.8 | 12.6 |

Table 7: Performance comparison of (final) PERM model against its variants to study the contributions of its components. The columns present the query structures and the overall average performance.

| Metrics | | HITS@3 | | | | | | | | | |
|---|---|---|---|---|---|---|---|---|---|---|---|
| **Dataset** | **Variants** | **1t** | **2t** | **3t** | **2∩** | **3∩** | **2∪** | **∩t** | **t∩** | **∪t** | **Avg** |
| FB15k-237 | 1t | .516 | .179 | .119 | .282 | .360 | .302 | .071 | .134 | .133 | .233 |
| | translations | .516 | .231 | .167 | .318 | .413 | .304 | .096 | .160 | .185 | .266 |
| | single | .511 | .282 | .212 | .359 | .486 | .296 | .126 | .207 | .235 | .302 |
| | average | .499 | .282 | .209 | .360 | .482 | .282 | .119 | .201 | .234 | .296 |
| | MLP | .510 | .285 | .212 | .363 | .488 | .293 | .125 | .208 | .238 | .302 |
| | (final) | .520 | .286 | .216 | .361 | .490 | .305 | .128 | .212 | .239 | .306 |
| NELL995 | 1t | .576 | .179 | .134 | .275 | .373 | .456 | .072 | .123 | .111 | .255 |
| | translations | .576 | .231 | .188 | .310 | .428 | .458 | .097 | .147 | .155 | .288 |
| | single | .571 | .282 | .239 | .350 | .504 | .446 | .127 | .190 | .197 | .323 |
| | average | .558 | .282 | .235 | .351 | .500 | .425 | .120 | .185 | .196 | .317 |
| | MLP | .570 | .285 | .239 | .354 | .506 | .442 | .126 | .191 | .199 | .324 |
| | (final) | .581 | .286 | .243 | .352 | .508 | .460 | .129 | .195 | .200 | .328 |
| DBPedia | 1t | .942 | .004 | N.A. | .781 | .734 | .775 | .129 | .600 | 0.00 | .496 |
| | translations | .942 | .006 | N.A. | .881 | .843 | .779 | .174 | .718 | 0.00 | .543 |
| | single | .934 | .007 | N.A. | 1.00 | 1.00 | .758 | .228 | .928 | 0.00 | .607 |
| | average | .912 | .007 | N.A. | .997 | .984 | .723 | .216 | .903 | 0.00 | .593 |
| | MLP | .932 | .007 | N.A. | .996 | .992 | .751 | .227 | .932 | 0.00 | .605 |
| | (final) | .950 | .007 | N.A. | 1.00 | 1.00 | .782 | .232 | .952 | 0.00 | .615 |
| DRKG | 1t | .560 | .202 | .130 | .302 | .396 | .373 | .106 | .172 | .165 | .267 |
| | translations | .560 | .260 | .183 | .341 | .455 | .374 | .143 | .206 | .230 | .306 |
| | single | .555 | .317 | .232 | .385 | .536 | .365 | .187 | .266 | .293 | .348 |
| | average | .543 | .317 | .228 | .386 | .531 | .347 | .177 | .259 | .291 | .342 |
| | MLP | .554 | .321 | .232 | .389 | .538 | .361 | .186 | .267 | .296 | .349 |
| | (final) | .565 | .322 | .236 | .387 | .540 | .376 | .190 | .273 | .297 | .354 |
| **Metrics** | | **Mean Reciprocal Rank** | | | | | | | | | |
| **Dataset** | **Variants** | **1t** | **2t** | **3t** | **2∩** | **3∩** | **2∪** | **∩t** | **t∩** | **∪t** | **Avg** |
| FB15k-237 | PERM-1t | .410 | .180 | .122 | .217 | .274 | .209 | .085 | .127 | .145 | .197 |
| | translations | .410 | .232 | .171 | .245 | .314 | .210 | .115 | .152 | .202 | .228 |
| | single | .406 | .283 | .217 | .277 | .370 | .204 | .151 | .197 | .257 | .262 |
| | average | .396 | .283 | .214 | .278 | .367 | .194 | .143 | .191 | .256 | .258 |
| | MLP | .405 | .286 | .217 | .280 | .372 | .202 | .150 | .198 | .260 | .263 |
| | (final) | .445 | .268 | .201 | .306 | .409 | .253 | .269 | .353 | .220 | .303 |
| NELL995 | 1t | .432 | .191 | .160 | .234 | .275 | .332 | .094 | .162 | .125 | .223 |
| | translations | .428 | .197 | .168 | .261 | .369 | .331 | .092 | .134 | .147 | .236 |
| | single | .425 | .241 | .213 | .294 | .435 | .322 | .120 | .173 | .187 | .268 |
| | average | .415 | .241 | .210 | .295 | .431 | .307 | .113 | .169 | .186 | .263 |
| | MLP | .424 | .243 | .213 | .298 | .436 | .319 | .119 | .174 | .189 | .268 |
| | (final) | .432 | .244 | .217 | .296 | .438 | .332 | .122 | .178 | .190 | .272 |
| DBPedia | 1t | .706 | .005 | N.A. | .665 | .541 | .564 | .169 | .791 | 0.00 | .430 |
| | translations | .700 | .005 | N.A. | .741 | .727 | .562 | .164 | .655 | 0.00 | .444 |
| | single | .694 | .006 | N.A. | .841 | .862 | .547 | .215 | .847 | 0.00 | .502 |
| | average | .678 | .006 | N.A. | .838 | .848 | .521 | .204 | .824 | 0.00 | .490 |
| | MLP | .693 | .006 | N.A. | .838 | .855 | .542 | .214 | .851 | 0.00 | .500 |
| | (final) | .706 | .006 | N.A. | .841 | .862 | .564 | .219 | .869 | 0.00 | .452 |
| DRKG | 1t | .416 | .173 | .116 | .254 | .341 | .269 | .100 | .157 | .157 | .220 |
| | translations | .416 | .223 | .164 | .286 | .392 | .270 | .135 | .188 | .218 | .255 |
| | single | .413 | .272 | .207 | .323 | .462 | .263 | .176 | .243 | .278 | .293 |
| | average | .404 | .272 | .204 | .324 | .457 | .250 | .167 | .236 | .276 | .288 |
| | MLP | .412 | .275 | .207 | .327 | .463 | .260 | .175 | .244 | .281 | .294 |
| | (final) | .420 | .276 | .211 | .325 | .465 | .271 | .179 | .249 | .282 | .298 |