# OpenReview forum: "Probabilistic Entity Representation Model for Reasoning over Knowledge Graphs"
_NeurIPS.cc/2021/Conference — NeurIPS 2021 Poster_

### Official Review · Reviewer_6UFN · 2021-07-16

**Rating:** 5
**Confidence:** 5

**Summary:**

This paper provides a novel approach to using embeddings to directly encode chains of reasoning over KBs with a particular focus on choosing a representation capable of handling unions. The proposed approach represents:

- Entities via Gaussian densities
- Relations (referred to as "translations") as an *action on mixtures of Gaussians*
- Intersection operations as a product of Gaussian densities
- Union operations as a self-attention weighted combination of Gaussian densities

The authors demonstrate an improvement over a variety of baselines for standard logic embedding evaluations as well as a COVID-19 drug repurposing dataset.

**Ethics Review Area:**

["I don’t know"]

**Limitations And Societal Impact:**

Throughout, the authors imply that their method of using Gaussians enables handling the union in a more principled way, however this claim is a bit dubious (aside from empirical evidence) and previous methods may also be extended in a similar manner. For example, Query2Box could have equally well interpreted their outputs (via min pooling as a result of DNF transformation) as representative of a multimodal score function. Perhaps the argument in favor of this approach is simply that the mixture of Gaussians is smoother, and thus trains better, but in fact a min (or max, in the PDF case) does seem more in-line with a union query, an observation which the authors ignore. One can simply view this as a difference in pooling functions (self-attention or min/max) and the loss by which a vector is compared to a query (i.e. Mahalanobis distance vs. weighted $\ell^1$).

**Main Review:**

### Overview

**Originality:** Are the tasks or methods new? Is the work a novel combination of well-known techniques? (This can be valuable!) Is it clear how this work differs from previous contributions? Is related work adequately cited?

**Quality:** The proposed model seems technically sound, there are a few issues with the mathematical formalisms which need to be clarified. The authors seems to have some misconceptions about previous work, however, which leads them to make stronger criticisms or claims about than are appropriate. (See below.)

**Clarity:** The submission is clearly written, however general editing and refactoring for clarity would improve it further.

**Significance:** The authors evaluate on a number of standard tasks which are of interest to the community, and provide convincing evidence that their model outperforms baselines.

---

### Specific Questions / Suggestions:

**L38:** The distance is typically not discontinuous. While for some methods the *gradient* may be discontinuous (eg. [0]), this typically isn't a problem for training. Even so, there are various ways to solve this, as have been developed in additional box embedding work (eg. convolution [1] or random latent variables [2]). This issue is also mentioned in the caption of Figure 2, and on line 62, but in each case the discontinuity is not really the problem. Query2Box, for example, uses a loss function with different weights inside the box or out. Similar to Gaussians, this will always assign some positive score to an entity, even outside the "box" which represents the query.

**L114-L116:** (Minor) It would be better to simply say that the translations are *actions* on Gaussian mixtures - they take in a Gaussian mixture and return a new one. There's not really a sense in which their parameters are, themselves, representative of the mean and std deviation of a Gaussian. (This criticism also applies to Query2Box, which characterizes projections as boxes when in fact they are actions.)

**L117-L124:** The product of two Gaussian PDFs is not Gaussian, rather it is *proportional* to a Gaussian. It is unclear to me whether you perform this rescaling or not - the definition of the intersection operation suggests that the result is always a normalized Gaussian, but the left-hand-side of equation (6) looks as though you are just multiplying PDFs, as does Figure 3(b). Depending on which you are performing, there may be an error in the derivation (which is fairly standard and may be moved to the appendix).

**L127-128:** In the self-attention mechanism, are you using the mean and covariance parameters or the PDFs of the Gaussians?

**L153:** Is the decomposed matrix $L$ diagonal? If so, I would suggest simply characterizing this fact as $\Sigma^{-1} = \text{diag}(\lambda)$, rather than the decomposition. If not, how does it only require $d$ parameters?

**Table 1:** FB15k-237, BQE actually performs best. In general, it seems BQE performs best for intersection queries. Do the authors have any intuition as to why this would be? (I would suggest that the reason has to do with the distinction of representing queries via random variables vs. densities - i.e. the BetaE paper represents queries as random variables, and the product of two Beta random variables is also a Beta random variable, whereas this paper proposes using the densities as the representation, and while the product of two Gaussian densities is also Gaussian the product of two Gaussian random variables is decidedly not Gaussian.)

[0] Ren, Hongyu, Weihua Hu, and Jure Leskovec. "Query2box: Reasoning over knowledge graphs in vector space using box embeddings." arXiv preprint arXiv:2002.05969 (2020).

[1] Li, Xiang, Luke Vilnis, Dongxu Zhang, Michael Boratko, and Andrew McCallum. "Smoothing the geometry of probabilistic box embeddings." In International Conference on Learning Representations. 2018.

[2] Dasgupta, Shib Sankar, Michael Boratko, Dongxu Zhang, Luke Vilnis, Xiang Lorraine Li, and Andrew McCallum. "Improving Local Identifiability in Probabilistic Box Embeddings." arXiv preprint arXiv:2010.04831 (2020).

**Time Spent Reviewing:**

6

---

> ### Author Response · Authors · 2021-08-09
> **Re: Official Review of Paper3863 by Reviewer 6UFN**
>
> We thank the reviewer for their constructive feedback. We are glad that the reviewer finds our paper well written and technically sound. Below we answer some concerns of the paper:
>
> **L38:** Query2Box adds the scalar weight to distinguish between $distinside$ and $distoutside$ to create a border for the box. However, the function is discontinuous because the difference between distances of points $\lim_{distoutside \rightarrow 0-}$ vs $\lim_{distoutside \rightarrow 0+}$ grows by a factor of $1-\alpha$. Hence, we try to solve this discontinuity in this paper with continuous Mahalanobis distance.
>
> **L114-L116:** While the operations are most definitely actions, we focus on translation as an operation to maintain consistency with previous literature in the area and also for better explanation of other operations; intersection and union; in the same context.
>
> **L117-L124:** The intersection is proportional to a Gaussian and this leads to information loss which leads to poorer performance in intersection queries, as seen in Table1. However, the trade-off is justified by the improved end-to-end framework that leads to better back-propagation for chains of different query types.
>
> **L127-128:** Yes, the self-attention used the mean and covariance parameters and not the pdf directly.
>
> **L153:** The $\Sigma^{-1}$ matrix is decomposed as $\Sigma^{-1}= LL^T$ where $L$ is a $d \times 1$ vector.
>
> **Table 1:** As given in L117-L124, the approximation of Gaussian intersections leads to this information loss.
>
> **Limitations And Societal Impact:** The problem with DNF over queries is that it requires the entire query to be analyzed together. This leads to scaling issues, which PERM solves by providing a closed solution to the querying process and hence, parts of the query can be analyzed in a smaller scope and merged in later steps, thus improving scalability.

---

> > ### Comment · Reviewer_6UFN · 2021-09-02
> > **Thank you for the clarifications, and some additional comments**
> >
> > Thank you for the various clarifications, most of my uncertainties have been addressed.
> >
> > As you subsequently acknowledged in response to another reviewer, the Query2Box loss is not actually discontinuous, rather it has discontinuous gradient. In practice, this is not a problem in practice for most optimizers (eg. Adam), and if this is the only issue with the model this could be easily solved by changing to a soft hinge function as opposed to (essentially) relu.
> >
> > Re: Limitations, while it is true that a closed solution to the querying process would solve scalability, you have not evaluated or demonstrated this additional scalability empirically - at most, considering the union over 2 elements, as in the original Query2Box model.
> >
> > With regard to both these points, I feel the fundamental limitation I mentioned in my initial review still holds: the model has fundamental incrongruencies in the choice of representation, and the aspects which lead to improved performance have not been sufficiently explored.
> >
> > Coupled with the comments of the other reviewers, I feel that a rating of 5 is appropriate for this work at this time, however I look forward to reading a refined presentation and analysis of this model in the future.

---

> > > ### Author Response · Authors · 2021-09-08
> > > **Re: Thank you for the clarifications, and some additional comments**
> > >
> > > Thanks a lot for the response.
> > >
> > > You are right, that the Query2Box loss is not discontinuous but actually non-smooth. However, the issue with the loss is not the gradient computation. The strict non-smooth borders of Query2Box make the answer representations very sensitive to the query borders and they cause incorrect ambiguity in answers that are nearby in the representational space. For e.g., assume that for a query, the result is a subset of all countries and the border needs to pass through a cluster of countries. If the border has a step factor $(1-\alpha)$, as in query2box, then the difference between countries inside and outside is exaggerated leading to false negatives, however with a smooth function like Mahalanobis distance, the difference between such clustered points would be better captured better. We show this in the empirical performance improvement given in Table 1.
> > >
> > > For scalability, we show in sections 3 and 4 that the query operations of translation, intersection and union can be organized (chained) in any possible combination to finally result in a gaussian mixture. This shows that complex queries are closed in the gaussian space. Hence, the queries can be modeled in a number of different combinations of single operations which inturn can be concurrently run on multiple devices.

---

### Official Review · Reviewer_TYBk · 2021-07-17

**Rating:** 5
**Confidence:** 4

**Summary:**

The proposed approach PERM is a knowledge graph embedding method that enables chain reasoning with first-order existential queries. In particular, it represents the query with a mixture of Gaussian distributions, constructed from intersection, union, and translation operations. Then, the result entities to the Gaussian mixture query are ranked using Mahalanobis distance. The evaluation shows that the proposed approach mostly outperform or comparable to the 4 baselines on 4 datasets.

**Limitations And Societal Impact:**

The impact is discussed in the paper. I agree with the authors, and I would consider the potential negative impact beyond scope of this work and it is where we also need to put more effort to prevent malicious knowledge graphs.

**Main Review:**

I think this can be an excellent paper for a few reasons, and the simple but new way to compute the first-order existential queries is empirically shown to outperform the baselines, and the analysis on how this approach works, and the ablation study are solid. Using a Gaussian distribution to represent a first-order logic looks sound in most cases, and it can have future research implications beyond this paper, considering typicality, etc. Although the paper is in general well-written, there is some information missing that can potentially make this paper very hard to understand for some out-of-field readers. The details are below.

Strengths
- S1: Solid experimental results showing the improvement on diverse datasets.
- S2: Additional experiments to show ablation study and visualization of a query and entities are helpful to understand the strength.

Weaknesses
- W1: The OR logic might not be best represented using the mixture of Gaussians because an entity can satisfy two predicates at once. For example, a person can be fully European and Canadian, if dual citizenship is allowed, and embedding an entity between two distributions might not be the best representation.
- W2: Only limited to first-order existential queries.
- W3: I understand the space is tight, but some key information about the training and some detail are missing. For example, the entire process is a bit vague, how to produce training data (not KGs, but queries), what is the dimensionality of the entities and so on because these are not mentioned or shown. For the overall process, an overview figure can be really helpful here. The overview becomes only slightly clearer in the algorithm in the appendix. The algorithm pseudo code might not be necessary, but some description should be there in the main text.

I like the paper in general, but I find the problem of missing some key information needs to be fixed before publication and make the paper self-contained as much as possible, if not citing an existing work. I'm willing to change the decision if it can be addressed even with the tight space.

**Time Spent Reviewing:**

3

---

> ### Author Response · Authors · 2021-08-09
> **Re: Official Review of Paper3863 by Reviewer TYBk**
>
> We thank the reviewer for their constructive feedback. We are glad that the reviewer finds our paper well written with solid experimental results. Below we answer some concerns on the weaknesses of the paper:
>
> W1: The union query is represented as the mixture of Gaussians. The illustration (Fig 2) shows how an entity embedding (a single point) would be best represented as a part of this Gaussian mixture. That is, Hinton as British Canadian should belong to $Europe \cup Canada$ and not between both of them as in boxes.
>
> W2: This paper focuses on tackling the first-order logic because a lot of problems fundamentally rely on this, such as, search and QA in KGs. However, we aim to extend our work to cover a wider range of problems.
>
> W3: The algorithm in the Appendix provides the overview of PERM. The implementation of PERM is also provided in the supplementary material for additional details. The methods for producing the training queries from the datasets is provided in the previous works in this field [7,10] and hence it is not repeated here. To maintain comparability, the dimension of the entity and relation is both set to 200, same as previous works in the area. However, we can include a figure to clarify the overview of the algorithm with additional information on the dimensionality.
>
> [7] Hongyu Ren*, Weihua Hu*, and Jure Leskovec. Query2box: Reasoning over knowledge graphs in vector space using box embeddings. In International Conference on Learning Representations, 2020.
>
> [10] Erik Arakelyan, Daniel Daza, Pasquale Minervini, and Michael Cochez. Complex query answering with neural link predictors. In International Conference on Learning Representations, 2021.

---

> > ### Comment · Reviewer_TYBk · 2021-08-23
> > **Union logic**
> >
> > Thank you for the clarification.
> >
> > Regarding your answer to W1, my point was that Hinton is 100% European as well as 100% Canadian. I think that is by the definition of a union query. He should be in the result if I query just Canadians, but he is far away from that.
> >
> > I believe this paper can be accepted at a major conference, but I think I need further clarification on this matter for that.

---

> > > ### Author Response · Authors · 2021-08-24
> > > **Re: Union Logic**
> > >
> > > Thanks a lot for the response.
> > >
> > > Yes, you are right that citizenship is a non-divisible attribute and should be treated as such. However, our model is query-centric and learns entity representations in a broader setting where Hinton needs to answer the queries Canada, Europe as well as $Europe \cup Canada$. This modeling technique also generalizes well over other datasets; such as drugs which need to be answers to protein interaction with multiple diseases and products which need to be answers for multiple categories. The task-specific weights assigned to a mixture component can be handled as a post-processing step. For e.g., we can threshold the mixture weights to be binary, i.e., lets say Hinton is learned as a mixture of $0.4 \times European+0.5 \times Canadian+0.1 \times Mexican$, we can normalize the weights to binary as $1.0 \times European+1.0 \times Canadian+0 \times Mexican$.
> > >
> > > Kindly, let us know if you need any additional clarification.

---

### Official Review · Reviewer_dgsE · 2021-07-17

**Rating:** 4
**Confidence:** 3

**Summary:**

The authors propose a Probabilistic Entity Representation Model (PERM) for learning query representations for chain reasoning over knowledge graphs. The model encodes knowledge graph entities as a Multivariate Gaussian density with mean and covariance parameters for encoding the semantic position and spatial query area of the entities. In particular, they utilize a mixture of multivariate Gaussian densities due to their intuitive closed-form solution for translation, intersection, and union operations. They claim that the closed-form solution for the operations allows them to solve complex queries by chaining them in a pipeline.

As contributions of the work, the authors provide Gaussian densities that can produce a closed-form solution to intersection and union and also a continuous distance function, which enables processing a chain of complex logical queries in an end-to-end objective function.
Furthermore, they indicate state-of-the-art results on logical query reasoning over multiple standard benchmarks and better drug recommendations for COVID-19 from the DRKG dataset. Finally, they declare that PERM is also interpretable as Gaussian embeddings can be visualized.


**Limitations And Societal Impact:**

The authors have appropriately addressed the limitations and potential negative social impact of their work in Section 7 of the paper.

**Main Review:**

The approach for utilizing a mixture of multivariate Gaussian densities to represent KG entities for learning query representations for chain reasoning is new in the field. The paper is well written and easy to follow. The methodology and claims are also supported. It is also nice that the code is provided.

However, I have concerns regarding the experiments part.
In the first experiment, the authors indicate SotA results compared to other approaches, and they also emphasize a comparison between Q2B and CQD models. Even though they visualize in Figure 2 why their model can be superior compared to union box queries, the authors do not explain why their model performs better for particular query types and worse for others (Table 1).

The ablation experiment looks solid. I only miss a mention of the benchmark for which results are reported in Table 2. Currently, it is not indicated in the table and also in the caption.

The case study for COVID-19 drug recommendation only indicates that the proposed model performs better than previous baselines and not why. It is similar to the first experiment. An error analysis here would be much more helpful. For instance, showing a particular set of queries or entities that the model fails to produce valid results would be much more useful.

The visualization experiment is also not convincing. Being able to visualize the entity and query in the latent space is interesting, but is only dedicated to one particular example. The visualization ability of Gaussian representations would match better with a solid error analysis or a case study with multiple examples covering different types of queries.

Overall, I find the work solid and interesting, but the experiments part weak. I believe it requires a revision to prove the superiority of the proposed model. For example, the authors need to provide why their approach works better for some query types and why not for others. Moreover, an error analysis and a case study for different query types, alongside the visualization ability, would provide more in-depth insights into the performance.


**Time Spent Reviewing:**

4-5

---

> ### Author Response · Authors · 2021-08-09
> **Re: Official Review of Paper3863 by Reviewer dgsE**
>
> We thank the reviewer for their constructive feedback. We are glad that the reviewer finds our paper well written and methodologically well supported.
>
> [Query type performance] PERM lacks performance on intersection queries because we use an approximation to model intersections. In all other cases, PERM shows higher performance. For intersections, we trade-off the performance to get an end-to-end architecture, so we can perform better back-propagation and integrate information from all query types.
>
> [Benchmark] As given in the caption, the results are to compare the different variants against the PERM(final) model.
>
> [COVID-19] In Table 3, drug recommendation results are meant to compare PERM’s results with the previous approaches in the area. The quantitative analysis of PERM on the DRKG drug dataset is given in Table 1.
>
> [Visualization] The visualization experiment is to show the potential explainability of PERM’s embeddings. The same method can be extended to any example/datapoint in the datasets.
>
> We prove the superiority of our model against previous works by quantitative experiments in Table 1 and Figure 2. Also, we show better performance in drug recommendation against previous models in Table 3. Furthermore, we show the final variant of PERM is better than all the other variants in Table 2.

---

### Official Review · Reviewer_HW1P · 2021-07-18

**Rating:** 5
**Confidence:** 5

**Summary:**

The paper proposes a new multi-hop reasoning algorithm on incomplete knowledge graphs. The method aims to embed complex logical queries and the chain reasoning steps as Gaussian mixtures. The benefit is that the model can provide extra flexibility in modeling union operations compared with previous embedding-based methods. The paper did extensive experiments, showed the proposed model outperforms prior models on the empirical performance and also demonstrated some visualizations and case studies. Overall the paper is well written and easy to follow. But the math/notations are somewhat messy (see below) and the paper is not that much well-motivated, especially for the specific design choices of the logical operators. The paper listed what they do but it would be better if the authors can justify why they want to have such design choices.

**Limitations And Societal Impact:**

Please check the main review section. I do not think it has any potential negative societal impact.

**Main Review:**

Please find my detailed comments and questions below.
1. Is the figure 2 a real example or is it a synthetic illustration? Why is the distance(hinton, union) smaller than distance(hinton, european) and also distance(hinton, canadian) in the Gaussian mixture model?
2. The description of the prior methods in the introduction has some errors. For example, line 38-41, “strict borders lead to ambiguity in the border case scenarios and a discontinuous distance function, e.g., a point on the border will have a much smaller distance if it is considered to be inside the box than if it is considered to be outside.” In fact, the according to Eq. 3 in [1], $\text{dist}_\text{box}(v;q)=\text{dist}_\text{outside}(v;q)+\alpha \cdot \text{dist}_\text{inside}(v;q)$. No matter the node on the border is considered insider or outside the box, the $\text{dist}_\text{outside}$ is always 0 and the $\text{dist}_\text{inside}$ is always the offset of the query box $q$. In other words, the distance function is continuous (piecewise linear function). Please justify your argument since I do not think border ambiguity is an issue for prior methods.
3. Regarding the other motivation “closed under union operations”, I think the beta query embedding [2] can also achieve closed union operation using the negation and intersection operations. Can you also clarify this point?
4. Do you also consider queries with both intersection and union operations? According to Eq. 1/2/3, it seems you do not. Since the paper emphasized chainable operations, it would be nice if the authors can discuss how to chain union and intersection in the same query.
5. The papers introduce how they design the translation, intersection, union operators in section 3. But they are not well motivated. Can you explain (1) why you ignore the covariance in the entity embedding when you calculate the distance between query and entity? (2) modeling the mean embedding as summation in the translation operator makes perfect sense to me, but why would you want to model the covariance as $\Sigma_e^{-1}+\Sigma_r^{-1}$?, what is the geometric intuition behind it? Does it shrink or expand the covariance matrix in general? More discussion can be super helpful. (3) you mention the weights for each Gaussian density is calculated using the self-attention mechanism, then are there any neural network modules with learnable parameters except for the embeddings to calculate the self-attention? I do not see any additional parameters in Eq. 7, and also the math is messy and has errors (see below). (4) Also why would you want to model the intersection of two \textbf{entity} density functions (line 118-120)? How would you define the intersection of two entities? If you mean the intersection of the two sets $\{e1\}\wedge \{e2\}$, then the results will be an empty set as long as $e1 \neq e2$. This is really confusing.
6. The math in Eq 7 is sketchy. Based on my understanding $\phi_i$ is a scalar or a vector (please clarify), and $\mathcal{N}(\mu_{e_i}, \Sigma_{e_i})$ is the pdf of the Gaussian distribution. Then why does $exp$ of a function returns a scalar ($\phi_i$)?
7. Also for the inline equation in line 138, it should be $p=\sum_{i=0}^n \mathbf{\phi_i} \mathcal{N}(\mu_i, \Sigma_i)$ instead of $p=\sum_{i=0}^n \mathcal{N}(\mu_i, \Sigma_i)$
8. How would you calculate $\phi_i$ in Eq. 8 (and also in Eq. 9-12)? Is it the same as Eq. 7? And does the $\phi_i$ is a self-attention over the \textbf{input} union query embedding or is it over the \textbf{output} union query embedding?
9. I have the same question in the Chain Intersection paragraph in Section 4. According to line 141-144, you are modeling the intersection between an entity and a query (a set of entities). But this intersection operation is trivial in the sense that the result will either be an empty set or a set of a single entity.
10. How would you model the intersection/union of two Gaussian mixtures?
11. Line 153-155, the paper mentions that “$d$ for $\Sigma^{-1}$”, so the $\Sigma^{-1}$ is factorized as the multiplication of a $L\times 1$ matrix and its transpose? Does it cause performance decrease since this has a large approximation error.
12. What is $\cap t, t\cap, \cup t$ queries? An illustration/example would be very helpful.
13. For the results in Table1, do you train PERM on all query types?
14. I still have questions on how PERM handles intersection operator and the self-attention mechanism, especially for ablation studies in Table 2. I wonder how PERM can answer queries with intersection if they are only trained on 1t queries. But this concern should be addressed if the authors can give me a detailed formulation of the self-attention mechanism and correct the math errors in Eq. 7.
15. Have you tried to measure the “uncertainty” of a query as in [2]? The idea is to calculate the entropy of the Gaussian mixtures and to find whether it shows any signal e.g., correlation with the number of answers of the query.

minor: you do not need to mention the distance function every time from Eq. 8 to Eq.12, since it is the same as in Eq. 5 to Eq. 7.

[1] Query2box: Reasoning over Knowledge Graphs in Vector Space Using Box Embeddings

[2] Beta Embeddings for Multi-Hop Logical Reasoning in Knowledge Graphs

**Time Spent Reviewing:**

6

---

> ### Author Response · Authors · 2021-08-09
> **Re: Official Review of Paper3863 by Reviewer HW1P**
>
> We thank the reviewer for their constructive feedback. Below we address the reviewers’ concerns on the various aspects of the paper:
>
> Figure 2 is a real example and the reason for the difference is that because Hinton belongs to both Europe and Canada, it lies in between the distributions and doesn’t belong to either of them, hence, its distance from the mean of both is high (>1). However, when we consider the mixture of these Gaussians, Hinton moves inside $Europe \cup Canada$ and hence the distance is much lower at 0.766.
> The function is discontinuous for border cases because the difference between distances of points $lim_{distoutside \rightarrow 0-}$ vs $lim_{distoutside \rightarrow 0+}$ grows by a factor of $1-\alpha$. Hence, we try to solve this discontinuity in this paper with continuous Mahalanobis distance.
>
> In beta embeddings, they still require the use of DNF to solve the union problem and also, their definition of negation of intersection covers all relations that are not in the intersection, while we need the relations specifically in the union and not the negation. Additionally, we compare our method against the beta embeddings in Table 1 and show that PERM performs better.
>
> The queries considered in this paper are for comparative study. However, given that the output of a union is a Gaussian mixture and that of intersection is a Gaussian pdf, we can use Eq.(9,10) and Eq. (7) to chain them with intersection and union, respectively.
> (1) Entities, here, have a dual nature. They act as Gaussian pdfs for query construction and only as mean points when acting as results. Hence, while calculating distance, we only consider entities as mean points. (2) $\Sigma^{-1}$ is a measure of how tightly the entities are clustered around the query mean. Hence, for translation, we would also like to loosen the cluster according to the relation using $\Sigma_1^{-1}+\Sigma_2^{-1}$, so we consider the covariance of both head and relation. (3) The embeddings and attention weights are both learnable parameters in the network.  (4) Intersection of two entities implies all their children (tail entities connected by relations) in common.
>
> In Eq.7, $\phi_i$ is an attention weight vector, for calculating the weights, we use $(\mu,\Sigma)$ as the embedding and calculate attention using only those vectors and not the pdf. $\phi_i$ is calculated in the same way for all the steps, but the attention weights are different for different operations. For unions, it is attention over the input embeddings.
>
> For intersection, the output will be a Gaussian distribution or a mixture depending on inputs. For answers, the entities are considered mean points and hence, their Mahalanobis distance from output Gaussian will decide their rank in the results.
>
> We model the intersection and union for two pdfs and  for a mixture and a pdf. To model with two mixtures, we can extend PERM as union over the first mixture with the operation over each component of the second mixture.
>
> Yes, the approximation leads to information loss. However, it is a necessary trade-off due to the computational expense of using the entire matrix. $\cap t,t \cap, \cup t$ are common operations in the area (also given in [8]). Yes, PERM is trained and evaluated on all the query types. That is another contribution, as the previous approaches couldn’t be trained on unions due to the non-existence of an end-to-end framework for back-propagation. Thanks a lot for the suggestion, we will add an “entropy” experiment in the appendix.
>
> The distance functions also work as the loss in the framework and hence, we add it for all the formulations for clarity and completeness.

---

> > ### Comment · Reviewer_HW1P · 2021-08-22
> > **Response**
> >
> > Thank you for the response.
> >
> > > The function is discontinuous for border cases because ...  grows by a factor of $1-\alpha$.
> >
> > This is not correct. The distance function of query2box is a continuous piecewise linear function. Please check the definition of continuous function [here](https://en.wikipedia.org/wiki/Continuous_function) and piecewise linear function [here](https://en.wikipedia.org/wiki/Piecewise_linear_function).
> >
> > > In beta embeddings, they still require the use of DNF to solve the union problem.
> >
> > As I said in the review, BetaE also has the De Morgan modeling of the union, no?
> >
> > > and also, their definition of negation of intersection covers all relations that are not in the intersection, while we need the relations specifically in the union and not the negation.
> >
> > Sorry I do not understand this sentence. Can you please clarify what you mean with an example? Thank you.
> >
> > > (4) Intersection of two entities implies all their children (tail entities connected by relations) in common.
> >
> > I also do not understand this sentence. Can you also give me an example? E.g., what is the intersection of Obama and Europe?
> >
> > > In Eq.7,  is an attention weight vector, for calculating the weights, we use  as the embedding and calculate attention using only those vectors and not the pdf.  is calculated in the same way for all the steps, but the attention weights are different for different operations. For unions, it is attention over the input embeddings.
> >
> > Thank you for the response, yes I know this is how you actually implement the method. I am saying the Eq. 7 tells a different story because you clearly define $\mathcal{N}(\mu, \Sigma)$ to be a density function (line 106, line 122), but what you actually do is to only use the vectors $\mu$ and $\Sigma$.

---

> > > ### Author Response · Authors · 2021-08-24
> > > **Re: Response**
> > >
> > > Thanks a lot for the response. Below, we provide some additional clarification on the points.
> > >
> > > >This is not correct. The distance function of query2box is a continuous piecewise linear function. Please check the definition of continuous function here and piecewise linear function here.
> > >
> > > Yes, you are right, the distance function of query2box is piecewise continuous and differentiable. But it is non-smooth. Hence, the difference between an inside point and outside point on the border is affected by a step factor of $1-\alpha$. This makes the answer representations very sensitive to the query borders. For e.g., assume that for a query, the result is a subset of all countries and the border needs to pass through a cluster of countries. If the border has a step factor, as in query2box, then the difference between countries inside and outside is exaggerated leading to false negatives, however with a smooth function like Mahalanobis distance, the difference between such clustered points would be better captured. We show this in the empirical performance improvement given in Table 1. We will edit the text in the final version to clarify this.
> > >
> > > > As I said in the review, BetaE also has the De Morgan (DM) modeling of the union, no?
> > >
> > > Yes, BetaE has De Morgan modeling. However, their experiments show that the DM modeling, on an average, does not perform as well as the DNF model (given in Table 1 of  [1]), whereas, our current Gaussian formulation preserves the closed union formulation while providing better performance (Table 1).
> > >
> > > >Sorry I do not understand this sentence. Can you please clarify what you mean with an example? Thank you.
> > >
> > > The DM formulation in BetaE is theoretically correct, however in practice, it has poor performance due to four sources of errors. Let us say we want to find $Obama \cup Europe$. For this, the BetaE-DM formulation would calculate $\overline{Obama}$, $\overline{Europe}$, then $\overline{Obama} \cap \overline{Europe}$ and finally $\overline{\overline{Obama} \cap \overline{Europe}}$, thus introducing four sources of errors. In our model, we do it in a single operation of Gaussian mixtures thus reducing the compounding from multiple sources of errors.
> > >
> > > > I also do not understand this sentence. Can you also give me an example? E.g., what is the intersection of Obama and Europe?
> > >
> > > Lets say there is a relation $meets$ with parent $Obama$ and another relation $politicans$ with parent $Europe$, then the intersection between Obama and Europe would result in all the European politicians that Obama has met such as Angela Merkel, David Cameron and others.
> > >
> > > > Thank you for the response, yes I know this is how you actually implement the method. I am saying the Eq. 7 tells a different story because you clearly define $\mathcal{N}(\mu,\Sigma) $ to be a density function (line 106, line 122), but what you actually do is to only use the vectors $\mu$ and $\Sigma$.
> > >
> > > Thanks a lot for the comment. We will clarify the equation further. Our point was to focus on the overall idea of attention in our model. However, we understand that the formulation may not be clear to the readers.
> > >
> > > Please let us know if you have further questions.
> > >
> > > [1] Hongyu Ren and Jure Leskovec. Beta embeddings for multi-hop logical reasoning in knowledge 311 graphs. In H. Larochelle, M. Ranzato, R. Hadsell, M. F. Balcan, and H. Lin, editors, Advances 312 in Neural Information Processing Systems, volume 33, pages 19716–19726. Curran Associates, 313 Inc., 2020.

---

### Decision · Program_Chairs · 2021-09-28

**Decision:**

Accept (Poster)

**Comment:**

The paper attempts to improve logical reasoning over knowledge graphs. In this regards, the authors propose a novel technique of  embedding complex logical queries and the chain reasoning steps as Gaussian mixtures. It is claimed that this allows for increased model expressibility and especially fits well for intersection. From empirical results, it seems the proposed method works well for certain types of queries. The reviewers found this work to be novel and interesting, but reached to a consensus that the reasoning provided do not corroborate with experimental results.  Also the reviewers found the presentation of some of the results hard to comprehend (e.g. the visualizations). Also it would be useful to understand and provide examples where the proposed model still fails. Thus, unfortunately I cannot recommend an acceptance of the paper in its current form, however the authors are strongly encouraged to make the fixes and resubmit to the next venue.

**Consistency Experiment:**

NeurIPS has a long history of experimentation. In 2014, NeurIPS ran an experiment in which 10% of submissions were reviewed by two independent committees to quantify the randomness in the review process. This year, we repeated a variant of this experiment to see how the quality of the review process has changed over time.  This paper was part of the experiment and was therefore assigned to two committees (consisting of reviewers, an Area Chair, and a Senior Area Chair) that reached independent decisions.  If both committees made the same recommendation, this recommendation was followed. If a single committee recommended acceptance, the paper was accepted (with the exception of a few cases in which the other committee identified what we considered a fatal flaw, e.g., an error in a key result).

This copy’s committee reached the following decision: **Reject**

The other committee assigned to the paper recommended **Accept (Poster)**.  You can find the other set of reviews, along with any follow up discussion with the authors here:
https://openreview.net/forum?id=AREHCsLy9oc